# A Conditional GAN for Generating Time Series Data for Stress Detection in Wearable Physiological Sensor Data

**DOI:** 10.3390/s22165969

**Published:** 2022-08-10

**Authors:** Maximilian Ehrhart, Bernd Resch, Clemens Havas, David Niederseer

**Affiliations:** 1Department of Geoinformatics, University of Salzburg, 5020 Salzburg, Austria or; 2Center for Geographic Analysis, Harvard University, Cambridge, MA 02138, USA; 3Department of Cardiology, University Heart Center Zurich, University Hospital Zurich, University of Zurich, 8091 Zurich, Switzerland

**Keywords:** time series GAN, generating measurement data, physiological sensor data, expert evaluation, machine learning, stress classification

## Abstract

Human-centered applications using wearable sensors in combination with machine learning have received a great deal of attention in the last couple of years. At the same time, wearable sensors have also evolved and are now able to accurately measure physiological signals and are, therefore, suitable for detecting body reactions to stress. The field of machine learning, or more precisely, deep learning, has been able to produce outstanding results. However, in order to produce these good results, large amounts of labeled data are needed, which, in the context of physiological data related to stress detection, are a great challenge to collect, as they usually require costly experiments or expert knowledge. This usually results in an imbalanced and small dataset, which makes it difficult to train a deep learning algorithm. In recent studies, this problem is tackled with data augmentation via a Generative Adversarial Network (GAN). Conditional GANs (cGAN) are particularly suitable for this as they provide the opportunity to feed auxiliary information such as a class label into the training process to generate labeled data. However, it has been found that during the training process of GANs, different problems usually occur, such as mode collapse or vanishing gradients. To tackle the problems mentioned above, we propose a Long Short-Term Memory (LSTM) network, combined with a Fully Convolutional Network (FCN) cGAN architecture, with an additional diversity term to generate synthetic physiological data, which are used to augment the training dataset to improve the performance of a binary classifier for stress detection. We evaluated the methodology on our collected physiological measurement dataset, and we were able to show that using the method, the performance of an LSTM and an FCN classifier could be improved. Further, we showed that the generated data could not be distinguished from the real data any longer.

## 1. Introduction

The rapid development of wearable sensors has led to a drastic increase in the availability and volume of physiological measurement data. At the same time, machine learning algorithms have gained momentum in recent years. This combination of new data sources and machine learning algorithms enables new human-centered applications in many research areas.

Originally, stress was introduced as the body’s response to environmental threats, triggering what is known as the fight or flight response [1]. When it comes to stress-related events, physiological measurement data provide insights into the autonomic nervous system [2]. The Autonomic Nervous System (ANS) in combination with the Hypothalamic–Pituitary–Adrenal (HPA) axis comprises the two main drivers of the body’s reaction to stress. Thus, physiological signals can be used to detect stress-related events. In particular, two reliable physiological signals to classify stress-related events are Galvanic Skin Response (GSR) and Skin Temperature (ST), because they are controlled by the ANS [3]. These signals can be measured using unobtrusive wearable sensors, which are easily integrable into everyday life and provide a large amount of data. This makes the development of new machine learning algorithms for physiological measurement data an attractive field.

However, machine learning algorithms typically require high-quality labeled data to deliver high-performance results. It is often difficult and costly to collect high-quality labeled physiological time series measurement data. This is due to several factors: Firstly, giving physiological values a label is a complex challenge that usually requires expert knowledge. Secondly, physiological measurement data usually need to be collected in strictly controlled environments to derive a labeled dataset. Lastly, it could be the case that the desired physiological event occurs only very rarely, leading to an imbalanced dataset. This is the case with our dataset, which is shown in Figure 1. As can be seen, using machine learning on small and imbalanced labeled physiological time series measurement datasets is challenging. Therefore, in machine learning, data augmentation is used to overcome the problem of small datasets by slightly changing the existing data samples to increase the variance in the whole dataset. Data augmentation is widely used for small datasets, especially in the domain of image classification tasks where images can be rotated, flipped, cropped, sheared, etc. [4]. However, it is not trivial to apply this technique to physiological measurement data and, at the same time, not changing the class of the measured physiological signal. Again, expert knowledge is required to correctly apply data augmentation to physiological measurement data. Previous success in the field of synthetic data augmentation encouraged us to use the Generative Adversarial Network (GAN) architecture to augment the existing physiological measurement dataset via synthetic data samples.

The GAN architecture is already well established when it comes to generating realistic synthetic image data. Because of their versatility, GANs have received much attention in many fields; one of these is medicine. Different use cases of GANs include cycle-GAN [5] in image-to-image translation, conditional GAN [6] (cGAN) to feed additional information into the GAN process, or style GAN [7], where the authors show the ability to control the generation process in more detail. However, the architecture was not only successfully implemented in the computer vision domain, but also in the time series domain. GANs are used to generate new data samples, hence they are proven to be a successful strategy to augment datasets for various classification tasks in order to improve classifier performance on small and imbalanced datasets [8]. To further improve the results, cGAN are used. All of the factors mentioned above inspired us to use the cGAN architecture to overcome the problem of the small and imbalanced physiological measurement dataset for machine learning.

In this work, we propose a novel methodology to tackle the problem of small and imbalanced datasets by generating synthetic labeled physiological time series measurement data, which are leveraged to improve the classifier’s ability to detect moments of stress. We replaced the recurrent discriminator with a Fully Convolutional Network (FCN), because in the early stage of our experiments, we obtained unstable results using a recurrent discriminator on our dataset. As a starting point, the recent success of FCN in time series classification [9,10] inspired us to utilize an FCN-discriminator in our cGAN workflow. The results from our experiments showed that the LSTM generator combined with a FCN-discriminator outperformed the recurrent discriminator. During training the cGAN, we used a combination of data preprocessing and a diversity term to overcome the problem of having an imbalanced dataset. After successfully training the cGAN, we augmented the dataset to be more balanced and increased the size of the dataset to improve the classifier score on stress detection. Our results show that it is possible to generate labeled physiological measurement data, which have the same underlying distribution as the real data, using the LSTM-FCN cGAN architecture.

## 2. Related Work

### 2.1. Detecting Stress-Related Events from Physiological Time Series Measurement Data

A wide variety of work has been performed in the field of stress recognition in recent years. Most of these studies deal with different datasets, methods of stress induction, and problem formulations, which makes it difficult to directly compare the algorithms. However, as [11] shows, different methods can be used to detect stress. One approach to evaluate stress is using physiological indicators such as GSR, ST, Heart Rate (HR), Heart Rate Variability (HRV) or the Interbeat Interval (IBI). These physiological measurements are mostly gathered using wearable sensors. The advantage of wearable sensors is that they can be unobtrusively integrated into everyday life situations, which makes it possible to classify stress in controlled laboratory environments and in real-world scenarios.

In the literature, different methods are proposed to detect moments of stress. For instance, Reference [12] proposed a rule-based algorithm to detect moments of stress using the physiological indicators of GSR and ST, collected from wearable sensor data. The authors developed a binary stress detection system, which distinguished between stress and non-stress states, and were able to transfer their results from the laboratory to a real-world environment. However, since the stress response usually varies from subject to subject, it can be challenging to find the perfect parameter set for tuning the rules.

In other studies, researchers used different physiological indicators and machine learning algorithms to detect stress. In particular, Reference [13] used HRV features and a Support Vector Machine (SVM) algorithm to detect stress based on a binary classification system. The work of [14] used the indicators of HR and GSR and applied the K-nearest neighbor classifier and Fisher discriminant analysis to detect stress in a real-world setting. All of these studies used wearable sensors to collect physiological measurement data and, subsequently, detected stress-related events.

Deep learning algorithms have shown great success in recent years for various classification tasks. However, limited studies have used deep learning algorithms for stress recognition. This could be because data acquisition in controlled environments is costly, and often, there is a limited number of participants, leading to less data availability, which, in turn, makes it difficult to train deep learning algorithms that perform well. Still, there are a few studies that have successfully used deep learning for stress detection. Reference [15], for example, used different machine learning and deep learning algorithms to detect stress using a neural network to classify stress from physiological measurement data. The authors of [16] used a Recurrent Neuronal Network (RNN), more precisely, a Long Short-Term Memory (LSTM) network, to detect stress from physiological measurements such as the IBI and GSR. The previous study is similar to the LSTM classifier architecture that we used to detect moments of stress, but uses a different combination of physiological stress indicators.

### 2.2. Conditional GANs for Time Series Data

The recent success of GANs in the domain of computer vision led to huge improvements in the field of machine learning. This has also led to different applications of the GAN architecture to time series data. For example, Reference [17] introduced T-cGAN to generate synthetic irregular conditional time series data. In this study, the authors first created 2D image spectrograms, which they subsequently mapped onto 1D time series samples. They used Convolutional Neural Networks (CNN) in the cGAN architecture. Another way of generating synthetic time series data was proposed by [18], who used a GAN and cGAN architecture based on LSTM for the generator and discriminator, instead of CNNs, to produce synthetic multidimensional medical time series data. This is the closest work to our cGAN architecture, as we also used a conditional LSTM-GAN. However, we additionally implemented a diversity term to overcome the problem of unstable cGAN training and replaced the discriminator with a FCN, to better extract features from the physiological signal.

### 2.3. Data Augmentation for Physiological Time Series Measurement Data

When it comes to classifying physiological time series measurement data, the problem of imbalanced and small datasets is often tackled with different data augmentation techniques [19,20]. Such techniques can be split into different categories.

One group of techniques is inspired by image data augmentation, where images are flipped, wrapped, or cropped to augment the dataset. In the work of [21], the authors propose techniques such as scaling, cropping, wrapping, or rotating the wearable sensor data to improve the classification performance for Parkinson’ disease monitoring. However, with this technique, it is important to ensure that the label of the respective data sample is not changed by augmenting the data, which is not always possible.

Deep learning is another method of data augmentation for physiological data classification tasks on small and imbalanced datasets. GANs can generate realistic synthetic data in various domains. This has been shown in the work of [22], who used a GAN to augment bio-signal data to improve the classifier performance on an imbalanced bio-signal dataset.

Conditional Generative Adversarial Networks (cGANs) modify the original GAN architecture, making it possible to add information to the generation process. For example, it is possible to add class labels as additional information to generate a labeled dataset. This opens up the possibility of using cGAN to augment training datasets for classifiers. Reference [23] used three different cGAN architectures and an adapted diversity term to augment a pathological Photoplethysmogram (PPG) dataset, to further improve a classifier. The authors of [24] proposed a method that generates synthetic physiological data with a cGAN to classify the arousal state of human subjects. This work is close to our objective, but there was very little description of the methodology used. Furthermore, we used a different neural network architecture to generate the physiological data for this study.

## 3. Methodology

The aim of the presented methodology is to capture the dynamic features of wearable physiological sensor data with the conditional Generative Adversarial Network (cGAN) framework to generate synthetic moments of stress. The physiological measurement data used in the methodology were collected in a controlled laboratory environment. After the data acquisition campaign, the data were prepared and preprocessed for usage in the cGAN workflow and classifier. In our methodology, we used the cGAN architecture with a diversity term [25], to produce more diverse labeled data. We used an LSTM for the generator and an FCN for our discriminator. After successful training of the cGAN, we were able to generate synthetic physiological measurement data related to the corresponding stress state label. This synthetically labeled physiological data were then used to augment the training dataset for the classification task. Given real physiological stress-related events and generated physiological stress-related events, the classifiers were able to distinguish the two states on our hold-out real test dataset. This section will describe the methods used in more detail.

### 3.1. Data Description

This section outlines the data acquisition campaign that we used to collect the physiological dataset. Our physiological time series measurement data can be divided into two classes. One class describes the moment when we induced a stress-related event in the controlled laboratory environment, which we call the Moment Of Stress (MOS). The other class, which we call the non-Moment Of Stress (non-MOS), represents every other moment during the laboratory experiment where we did not trigger a stressor. Every session lasted at least 12 min for each participant. To reduce the complexity of the sequences and to label our dataset, we decided to use a sliding window of 16 s and only used every 16 s interval for the non-MOS, to balance the ratio between MOS and non-MOS. This process is shown in more detail in Figure 2. If this 16 s interval fell into the MOS category, we dropped that sequence, because we did not want an MOS sequence labeled as non-MOS. For the MOS class, we used every sample, where we introduced a stress event via an air horn sound. We took one second before the stress event and 15 s after the stress event to obtain a 16 s sequence. The length of 16 s accounts for the 1.5–6.5 s delay [1], which is the time taken from the onset of the stressor to the rise of the GSR, the next 2–5 s, which is the time taken for the GSR to rise, and the 1–10 s recovery time for the GSR to return to its original state. Concerning the duration of an MOS, the change in the ST signal must be accounted for. The ST behaves dependently onthe GSR. After a 3 s GSR rise, the ST starts to fall for at least 3 s after a moment of stress occurs. However, this drop can last up to 6 s in extreme cases, as our data show. The average duration of an MOS is 10 s [12], but to capture all variations in our data, 16 s is the optimal time frame. After the downsampling process, we obtained 280 MOS samples and 1282 non-MOS samples of 16 s each in total.

### 3.2. Data Acquisition Campaign in a Controlled Laboratory Environment

As for any machine learning algorithm, training data are needed. When it comes to physiological measurement data related to stress events, to the best of our knowledge, no comparable uniform dataset exists. For this reason, we conducted a data acquisition campaign in a controlled laboratory environment to acquire real-world stress data. The Autonomic Nervous System (ANS) controls many physiological reactions of the body. The ANS is also responsible for the fight or flight response. As such, measuring physiological signals can give us unambiguous insights into the ANS and, consequently, the stress level of a subject. For this reason, we set out to measure the body’s physiological reaction to a stressor in a controlled laboratory environment to generate a gold standard for moments of stress. However, we must acknowledge that there is no objective gold standard because we cannot determine what people really feel, and a physiological reaction to an emotional–psychological process is not a perfectly reliable indicator of stress. Therefore, our primary goal for the data acquisition campaign was to minimize uncertainty and define the occurrence of moments of stress as precisely as possible. In the next section, this procedure is described in more detail.

#### Setup

We collected the physiological data used in this study with wearable sensors. These sensors transmitted the data to our e-diary app via Bluetooth, and the data were saved to SQLite files. We used the Empatica E4 [26] wearable sensor to measure GSR and ST. The E4 is a high-grade wearable sensor, designed for research and clinical trials. With the E4, we sampled both GSR and ST with a rate of 4 Hz. The E4 sensor measures GSR with a resolution of 900 pico Siemens and within a range of 0.01 to 100 μS. Concerning ST, the E4 measurement has a resolution of 0.02 °C, and the accuracy within the 36–39 °C range is ±0.2 °C. The maximum possible measurement range of ST is within −40 to 115 °C.

A total of 35 subjects participated in the data acquisition campaign. The age of the participants ranged from 18–55, and the gender of the participants was evenly distributed with 17 male and 18 female participants. The participants were split into subgroups of 5 people per session. The sessions were always held from 10 am to 2 pm on three different days. The invitation to the data acquisition campaign was mainly sent via the internal student representation server. Some participants were also invited by e-mail or via personal contact. After a short introduction outlining the experiment’s procedure, the participants were placed in a room and equipped with E4 sensors. To prevent bias as much as possible, the participants were asked not to consume stimulant drinks prior to the experiment. In addition, all participants were healthy and not taking any medications. During the experiment, each participant was seated on a chair facing the wall to prevent interference with each other. The chairs were arranged in a circle, and the noise source was placed in the middle of the circle. During the experiment, ten stress moments were induced via an air horn sound. This sound was played with a “JBL charge 3” music box. The stressors were induced at random time intervals, but the time between stressors was generally longer than 60 s. With the exact time of the air horn sound, we could locate the change in the physiological signal and, thus, the reaction of the body to the stimuli.

### 3.3. Data Processing

Next, we had to preprocess and transform the sensor datasets from the controlled data acquisition campaign in Section 3.2 to be processed by the LSTM-FCN-cGAN model. In this section, the necessary steps are described. Figure 2 shows the process.

Although our sensors collected several physiological measurements, the Galvanic Skin Response (GSR) and Skin Temperature (ST) were the most relevant for our study, due to the instant and reliable reaction of these variables to the stress stimuli [12,27]. The GSR signal was divided into two different categories: the Skin Conductance Reaction (SCR), which occurs directly after a stressor, and the Skin Conductance Level (SCL), which describes the baseline level of GSR. For this study, we were interested in the SCR because we wanted to detect instant reactions (MOS). First, a first-order Butterworth low-pass filter with a cut-off frequency of 1 HZ was applied to cut off fluctuations and noise from the GSR signal [12,28], caused by, for example, movement. This cut-off frequency was found using a fast Fourier transformation, which showed noise in the frequency spectrum of the signal and a visual evaluation of the filter signal. To separate the SCR from the SCL, a first-order Butterworth high-pass filter with a frequency of 0.05 HZ was applied to the signal, as proposed by [12,28]. To reduce noise in the ST signal, we first applied a second-order Butterworth low-pass filter with a cut-off frequency of 0.1 HZ and then a second-order high pass filter with a cut-off frequency of 0.01 [12,29], to remove the small fluctuations in the signal. After that, both signals were downsampled to 1Hz to further smooth the signal. The downsampling process involves calculating the average for each one-second window. The sequences of data per participant were up to 15 min long. To reduce the complexity in the data, we used a sliding window of 16 s, as described in Section 3.1. For the non-stress moments, 16-second sequences were used in which no stressor occurred, and we labeled these sequences with a zero. After this downsampling and labeling process, we had 16 s sequences in the form of matrices $X\in R^{n\times t\times d}$, where *n* denotes the number of sequences we obtained after preprocessing, *t* the number of time steps of each sequence, and *d* the number of features, which in our case stands for the number of preprocessed signals such as GSR and ST. The 3D matrix *X* was than pushed through our LSTM generator and through the FCN discriminator. The labels are vectors $c\in {\left\{0,1\right\}}^{n}$.

#### 3.3.1. Train-Test Split

Before feeding the data into our workflow, we split them up into a training dataset (80% of data) and a testing dataset (20% of data) [30]. This way, we can counteract the possibility of the generator memorizing the dataset, which means that the cGAN would not learn any valuable features. The training dataset was used to train the cGAN, and the testing dataset was used to evaluate the methodology. Therefore, we could test the methodology using data that neither the classifier nor the cGAN had seen. The data were split into training and testing datasets by picking randomized stress sequences for each participant. This helped us reduce bias because the stress levels decline during the experiment, since people show stronger reactions at the beginning of the experiment than later on. This was shown in our experiment and in the experiment conducted by [12] and can be attributed to the fact that the subjects get used to the stressor after time.

### 3.4. GAN Architecture and Model Training

The core of the proposed methodology is the conditional GAN [6] (cGAN) model. The cGAN workflow is shown in Figure 3. The model is able to generate labeled synthetic moments of stress with temporal signatures that are statistically similar to the data we acquired in the controlled laboratory environment. In the GAN [31] model, there are two networks competing to optimize one another in a min–max game. The so-called “generator” tries to create samples that are similar to the samples drawn from the real distribution to fool the discriminator. The discriminator then has the task of distinguishing between real samples and samples from the generator. Because we have a feed-forward neural network, we can use backpropagation to update the weights of each network in the GAN model until they reach the desired equilibrium point. In an ideal scenario, the discriminator can no longer distinguish between real and fake samples, and we have a generator that can produce synthetic samples following the same distribution as the ones drawn from the real one.

#### 3.4.1. Temporal Fully Convolutional Networks

In our architecture, the temporal convolutional layers are used to extract features from the time series signal. The convolutional layer for the discriminator is chosen, because in our experiments, we saw that the FCN discriminator outperformed the recurrent discriminator. This indicates that the convolutional network, especially the FCN, provides the generator with better gradients during training. Therefore, 1D filters were applied to capture the changes in the signal according to the different classes of physiological signals. As described in [32], the filters for each layer are learned by the weight tensor *W* and biases *b*. Therefore, we have:(1)$$\begin{array}{c} {\tilde{A}}^{\left(l\right)}=f\left(b_{k}^{\left(l\right)}+\sum_{i=1}^{d}\langle W_{i,k}^{\left(l\right)},A_{k}^{\left(l-1\right)}\rangle \right) \end{array}$$
where *l* denotes the index of the layers, *d* the length of each filter, ${\tilde{A}}_{k}^{l}$ denotes the activation at layer l at the *k*th neuron, and *f* is the activation function used.

#### 3.4.2. LSTM Network

The main goal of our framework is to learn the representative features in GSR and ST related to their label in a fixed time frame. To achieve this, we used an LSTM neural network [33] as the generator, like in the work of [18], but we replaced the discriminator with an FCN. The LSTM is part of the recurrent network family, which predict the next step in time, leveraging the present and previous states in time. This makes them well suited to process sequential data. In this study, we used the LSTM over a simple Recurrent Neural Network (RNN) because RNNs suffer from vanishing or exploding gradients [34]. LSTMs are more robust against vanishing or exploding gradients because of their different gates, which determine the information that is removed or used to update the current cell state. An LSTM cell has different layers: an input gate layer $i_{t}$, a forget gate layer $f_{t}$, an update gate layer $c_{t}$, and an output gate layer $o_{t}$. The complete cell state structure is shown in Figure 4, and the calculation procedure can be described as follows.

First, the current memory cell is calculated. $W_{c}$ denotes the matrix storing the weights, $h_{t-1}$ the last hidden state and $b_{c}$ the current bias.
(2)$$\tilde{c_{t}}=tanh\left(W_{c}\cdot \left[h_{t-1},x_{t}\right]+b_{c}\right)$$

Next, the input gate layer decides, via a sigmoid activation function, which information to keep from the input signal. In our case, we have a matrix *X*, as described above, which contains sequences of physiological data. Within this context, $W_{i}$ denotes the input weight matrix and $b_{i}$ the input bias.
(3)$$\begin{array}{c} i_{t}=\sigma \left(W_{i}\cdot \left[h_{t-1},x_{t}\right]+b_{i}\right) \end{array}$$

The input for the forget gate layer is the previous output state concatenated with the current input vector. A sigmoid activation function is used to determine which historical information is relevant and which information should be removed. Within this context, $W_{f}$ denotes the forget weight matrix and $b_{f}$ the forget bias.
(4)$$\begin{array}{c} f_{t}=\sigma \left(W_{f}\cdot \left[h_{t-1},x_{t}\right]+b_{f}\right) \end{array}$$

Then, the input state is combined with the current cell state and the forget gate layer is combined with the historic cell state to update the next cell state, where ⊙ denotes elementwise multiplication.
(5)$$\begin{array}{c} c_{t}=f_{t}⊙c_{t-1}+i_{t}⊙\tilde{c_{t}} \end{array}$$

The output state layer controls the information, which is passed to the next hidden state. Here, $W_{o}$ denotes the output weight matrix and $b_{o}$ the output bias.
(6)$$\begin{array}{c} o_{t}=\sigma \left(W_{o}\cdot \left[h_{t-1},x_{t}\right]+b_{o}\right) \end{array}$$

The last step is the output gate layer, which decides what information gets to the next hidden state. Therefore, the current cell state is pushed through a tangent hyperbolic (tanh) activation function and the gets combined with the output state.
(7)$$\begin{array}{c} h_{t}=o_{t}⊙tanh\left(c_{t}\right) \end{array}$$

#### 3.4.3. Conditional GAN

Instead of using the standard GAN framework, we utilized the conditional Generative Adversarial Network (cGAN) [6] to generate labeled data. The conditional information is vector-based, as described by *c*, indicating whether the samples are from the distribution of stress moments or non-stress moments. We then concatenated the matrix *X* with the labels $ystress_{n}$ before feeding them into the LSTM. Before we concatenated the two inputs, we fed $ystress_{n}$ through a 2D categorical embedding layer and then upsampled it to the shape of *X* with a linear dense layer. Through the embedding layer, the neural networks learn a mapping between class label information and sequences, to further be able to better control the conditional generation process. The structure of the conditional generator and discriminator is shown in Figure 5.

#### 3.4.4. Model Training

The cGAN architecture, as described in Section 3.4.1 and Section 3.4.2, consists of two networks, which are simultaneously optimized via backpropagation. For this, the min–max adversarial loss function is used, whereby the discriminator tries to maximize the log probability of labeling real and fake data correctly, and the generator tries to minimize the probability of being classified as fake log(1−D(G(z)), where the latent space *z* is sampled from a Gaussian distribution $N(\mu ,\;\sigma^{2})$ [35]. As proposed in the original GAN paper [31], this leads to the following objective function:(8)$$\begin{array}{c} \underset{G}{min}\;\underset{D}{max}=E_{x,y}\left[log\left(D\left(x|y\right)\right)\right]+E_{z,y}\left[1-log\left(D\left(G\left(z|y\right)\right)\right)\right] \end{array}$$

During the optimization process, Equation (Equation 8) often results in mode collapse, which means that many samples out of the latent space map to the same generated sample. This results in a dataset with less diversity. To counteract this problem, the diversity term was introduced by [25], to simply regularize and penalize the generator for producing the same samples. The diversity term is defined as:(9)$$\begin{array}{c} \underset{G}{max}f\left(G\right)=E_{z_{1},z_{2}}\left[\frac{∥G\left(z_{1},y\right)-G\left(z_{2},y\right)∥}{∥z_{1}-z_{2}∥}\right] \end{array}$$
In that way, the basic idea is, if two samples are different, but the generated sequences are the same, the term is 0. Therefore, the generator’s objective is to maximize this term. This results in the following new objective function:(10)$$\begin{array}{c} \underset{G}{min}\;\underset{D}{max}\;f\left(G,D\right)-\lambda f\left(G\right) \end{array}$$
where λ is a hyperparameter, which describes the importance of the term in Equation (Equation 10) and ‖ denotes a norm. In our experiment and in [25], 8 was proven to be a good value.

In the generator, a stacked LSTM with 16 hidden units per layer is used to generate physiological signals. Before the two LSTM-layers, there is the 2D categorical embedding layer, which is mentioned in more detail in Section 3.4.3, and a linear layer to learn the label of the stress data during adversarial training. In the generator, the mapping from the random space is performed via a dense layer using a Leaky ReLU activation function. After that, the LSTM layer group is applied. The output is fed through a linear activation instead of tanh, because, during our experiments, scaling the values to a range of −1 to 1 did not work out. The final output of the generator has the shape of the matrix *X* mentioned in Section 3.3.

The discriminator is the FCN proposed by [10]. As mentioned above, the label information is first pushed through the 2D embedding layer and then gets upsampled via a dense layer, before it gets concatenated with the input sequences. To learn the relevant features of the physiological signal, there are three temporal convolutional network blocks followed by batch-normalization and a ReLU activation function, which results in the following equations:(11)$$\begin{array}{c} y=Conv1D \end{array}$$(12)$$\begin{array}{c} n=BatchNorm(y) \end{array}$$(13)$$\begin{array}{c} h=ReLU(n) \end{array}$$

The filters per layer are {32,64,32}, and the kernel size per layer was set to {8,5,3}. After the three convolutional blocks, the resulting feature maps go through a global average pooling layer. The output from the global average pooling is then pushed to the sigmoid activation function, which outputs a scalar value in the range of 0 to 1 for the sequence, indicating whether it is real or fake. This results in the shape of $R^{n\times 1}$ for the output of the discriminator.

For the optimization process, we used the Adam optimizer [36] with a learning rate of 0.0002 and a beta value of 0.5 [37] and trained it for 1750 epochs. A batch size of 32 was used to ensure stable training.

### 3.5. Evaluation

Evaluating GANs in the time series domain is not trivial; hence, we decided to use several methods for our evaluation to be able to make a statement about the quality of the generated data. For this reason, we used the following criteria:Discriminability of synthetic and real sequences, which means that we want to show that our generated data are no longer distinguishable from real data samples;Variety of synthetic sequences, where we want to show that our generated data cover as many different modes of our real dataset as possible;Quality of the generated sequences, where we want to show that the generator captured the dynamic features of our real dataset.

#### 3.5.1. Visual Evaluation

First, as with image data, people can evaluate the quality of samples visually [38] to discriminate between real and fake samples. In this case, the results should be near chance level, which means that there is no difference between real and fake samples. The drawback of this approach is that only a few people are able to evaluate moments of stress. Therefore, we conducted an expert evaluation experiment, where different experts from the human sensing field, such as cardiologists, evaluated our generated MOS and non-MOS. In their evaluation, the experts had to decide whether the sample was artificially generated or real and, in the second step, whether a sample was an MOS or non-MOS. We used a simple form to facilitate these expert evaluations. Another method to visually compare the quality of the generated samples is the t-sne visualization [39]. With the help of the t-sne visualization, higher-dimensional data can be displayed in a 2D map. This was performed in a way that similar objects are displayed closer together than dissimilar ones on the map. Gradient descent was used to reduce the Kullback–Leibler divergence between the 2D pointwise distribution and the higher-dimensional data distribution. For the above reasons, it can be used to diagnose that the synthetic data match the real data, no mode collapse happens, and whether some modes of the dataset are omitted during training. One more method for visual evaluation is looking at the temporal properties of each generated MOS. Here, we can observe whether the generator can reproduce the proposed rules that characterize an MOS, which was described in Section 3.1.

#### 3.5.2. Statistical Evaluation

The samples produced by the cGAN architecture are of good quality if there is no difference between generated data and real data. To evaluate this similarity, we used the Classifier Two-Sample Test (CTST) proposed in the paper [40]. In this approach, a binary classifier is trained to distinguish samples belonging to the synthetic dataset $\tilde{S}$, from the real dataset *S*. The sampling process is performed by randomly picking samples from the real dataset and from the synthetic dataset, with $|\tilde{S}\left|=|S\right|$. The same procedure is used for the test dataset. After the training process, the classifier is then evaluated on the hold-out test dataset. If the test accuracy is near the change level, there is strong evidence that the distributions are the same. The outcome of this test can also be used to diagnose which failure occurred during cGAN training and for model selection during training, as stated in the paper [40] and as our experiments showed. The classifiers predictions for the individual samples are the indicators where the distributions differ. The CTST can also help to monitor the evolution of the cGAN model over time towards a theoretical equilibrium point. Thus, the stability of the parameters can be better assessed and, for example, early stopping can be applied. Another method to evaluate the quality of the generated samples is the training on generated data and testing on real data approach [18], with the assumption that if the classifier, trained with synthetic data, can correctly classify real data that the classifier and the cGAN have never seen, then the synthetic data will be of high quality and variety. To further underline the above criteria, we used our trained generator to augment the training dataset with synthetic data to increase the classifier’s performance and compare it to the real score baseline. If the classifier score increases, it can be assumed that the data contain the information of the original dataset and the generated samples have reasonable variance [8]. For both methods, we used a stacked LSTM and an FCN classifier, which is described in Section 3.5.3, to distinguish between MOS and non-MOS.

#### 3.5.3. Classifier Architecture

We utilized two different classifier architectures to further evaluate our methodology. Therefore, we used a temporal FCN and a stacked LSTM network. The LSTM classifier consists of two layers with 50 hidden units each. The temporal FCN consists of three layers. Each layer is composed of a batch-normalization followed by a ReLU activation. Regardless of whether a sample is an MOS or non-MOS, the output probability of both networks comes from the sigmoid output activation, with the formula:(14)$$\begin{array}{c} \phi \left(z\right)=\frac{1}{1+e^{-z}} \end{array}$$

Each model was trained using the binary cross-entropy loss function and optimized via the Adam optimizer. In the LSTM, the learning rate of the Adam optimizer was set to 0.001, and in the FCN, the learning rate was set to 0.0001. The formula to minimize the binary cross-entropy is given, where θ denotes the output from the activation function, y the true label, and x the output probability from the sigmoid activation function.
(15)$$\begin{array}{c} L\left(\theta \right)=-\frac{1}{N}\sum_{i}^{N}\left[y^{i}log\left(h_{\theta }\left(x^{i}\right)\right)+\left(1-y^{i}\right)log\left(1-h_{\theta }\left(x^{i}\right)\right)\right] \end{array}$$

## 4. Experiments and Results

In this section, we show the performance of the proposed methodology on our collected physiological stress dataset. All methods used were described in the previous section. The obtained results were divided into visual results and statistical results.

### 4.1. Generated Moments of Stress

After the generator is successfully trained, we can apply random Gaussian noise and class labels to the generated MOS and non-MOS samples. These generated samples can be visualized. As shown in Figure 6, the cGAN captured the latency from the stressor to the rise in GSR, the GSR spike, and the recovery of the signal. Regarding skin temperature, the generator learned that a short drop in skin temperature occurs after the skin conductivity increases. The samples indicate that the generator has learned to generate different variations of MOS samples that match the time–frequency characteristics of the real data. This can be observed in more detail when looking at the various latencies between the stressor and the onset of the peak, different amplitude levels, variations in the GSR rise time, and different ST curves. Concerning the non-MOS samples, the generator learned to produce flat curve samples with little fluctuations or samples with multiple peaks, which indicate noise. In general, for both classes, there is no noticeable difference between the real and synthetic samples. The generated samples, shown in Figure 6, are the same samples from the same generator, which were fed into our classifier to augment the dataset.

### 4.2. t-sne Results

The two plots in Figure 7 demonstrate the t-sne results for the MOS and non-MOS class. Each data point in the plot consists of 16 s GSR and ST signal features, which were mapped to a 2D plane. The red points are from the generator, and the blue points are from the real dataset. Points, which are closer together on that plane, have a higher probability to be similar. This means that the real and the synthetic data points should overlap in the graphic. As the data points show in the plot, the generated data points from both classes correspond to the real points, which fulfill the above-described criteria. In addition, there is a neighbor to most points, which means that no mode was omitted during training. This also indicates that mode collapse did not happen. The 2D values of that plot were created via the t-sne implementation of sklearn [41].

### 4.3. Expert Assessment Experiment

The experiment was designed around the criterion of discriminability, mentioned in Section 3.5. For this purpose, 5 experts in the field of human sensing from various domains of physiology were recruited to classify the generated and real data via a web-form. We used 30 real MOS, 30 generated MOS, 50 real non-MOS, and 50 generated non-MOS. The real and generated sequences were randomly sampled. In addition to the classification between generated and real, a classification between stress and non-stress was also investigated and compared with our results from Train on Generated, Test on Real (TGTR) classification, which is shown in Table 1. Because of the lack of experts that are able to classify stress moments and the resulting low participant rate in the experiment, we therefore utilized the mean value of all the participants’ classification performances. The answer options were presented in a form, where first, the question about generated and real had to be answered and, then, a distinction between MOS and non-MOS had to be made. The physiological signal GSR and ST that had to be evaluated were combined and displayed in one plot. For each participant, the results were stored in a CSV file. As proposed early on, the results of distinction between generated and real showed that our generated data were visually the same as the real data, because of the score, which was near the chance level, namely 0.4575 accuracy over all samples. To further evaluate our generated data, we utilized the classification between stress and non-stress. There, real and generated stress moments were almost equally classified with a difference of 1 misclassified MOS sample, which is indicated in the recall score of around 0.74 (real) and 0.7733 (generated). All sequences are the generated and real samples combined. All results are shown in Table 2 and Table 3.

### 4.4. Classifying Moments of Stress

All our models were implemented with TensorFlow and Keras [42]. To evaluate our methodology, we used different evaluation metrics to measure the performance of a binary classifier. The first one is recall, which describes the true positive rate. This metric tells us how many of the MOSs the classifier can correctly classify. The next score we used is precision, which describes the model’s performance in predicting actual positive samples as positive. It is calculated as the ratio of true positives among all positive classified samples. The F1-score describes the balance between the recall and precision.

Accuracy shows the ratio of correct predictions among all the sequences in the dataset, which means it returns all true positives and true negatives. Below are the formulas for the above-mentioned metrics, where TP denotes True Positive, TN denotes True Negative, FN denotes False Negative, and FP denotes False Positive.
(16)$$recall=\frac{TP}{TP+FN}$$
(17)$$precision=\frac{TP}{TP+FP}$$
(18)$$f_{1}=2\cdot \frac{precision\cdot recall}{precision+recall}$$
(19)$$accuracy=\frac{TP+TN}{TP+TN+FP+FN}$$

The classifier model that we used in our experiment was described in Section 3.5.3. We used the real data classifier score as a baseline, which we then compared to the “Train on Generated Test, on Real”(TGTR) classifier score and to the Data Augmentation (DAug) classifier score. Furthermore, we compared our model with the state-of-the-art techniques in the field of time series generation with GANs and cGANs:Recurrent Conditional GAN (RCGAN) [18], where two recurrent networks as generator and discriminator are used. There is also the possibility to add label information in the generation process.TimeGAN [43] is a GAN framework for generated time series data. Different supervised and unsupervised loss functions are combined to generate the data.

After the train-test split, the training dataset size was 1222 sample sequences, consisting of 196 MOS and 1026 non-MOS, whereby each sequence was 16 s long. All three methods were tested on the same dataset, which came from the train-test split described in Section 3.3.1. The testing dataset consisted of 340 samples, with 84 MOS and 256 non-MOS, with each sample lasting 16 s.

#### 4.4.1. Train on Generated, Test on Real

In Table 1, we compare the baseline score, the RCGAN [18] score, and the TimeGAN [43] score against our approach. The results showed the optimal weight composition we found during the training. To train the classifiers, we used only synthetic MOS and synthetic non-MOS generated by the methodology. We used the same ratio of MOS and non-MOS samples as in the real training dataset. Each classifier was then evaluated on the testing dataset, which consisted of real data only. We generated a dataset out of a fixed random seed with the same size as the real one. The baseline results of the classifier were a precision score of 0.8542 for the FCN and 0.8654 for the LSTM. The recall score of the FCN was 0.4881 and for the LSTM 0.5357. The F1-scores of the baseline classifiers were 0.6212 for the FCN and 0.6618 for the LSTM. If we compare the baseline results to our TFTR score, where the classifier precision scores were 0.7719 (FCN) and 0.76 (LSTM), the recall scores were 0.5238 and 0.6786, respectively. It can be observe that there was improvement in the recall for both models and a slight decrease in precision. Simultaneously, the F1-score increased for both models in comparison to the baseline real data classifier score. These results suggest that the generator captured the important features from the real dataset and was therefore able to produce valuable results.

#### 4.4.2. Data Augmentation Results

The Data Augmentation (DAug) results are in Table 1, and it depicts the optimal weight constellation we found during training. The number of synthetic samples used for data augmentation was a hyperparameter because, at a certain point, adding more synthetic samples to the classifier no longer improved the score. During our experiment, we found that 800 synthetic samples per label comprised the best possible number to augment the dataset, considering training speed and classifier score. As shown in Table 1, we compared the RCGAN [18] and the TimeGAN [43] with our LSTM-FCN cGAN. There, the data augmentation method with our approach performed the best and improved the recall score and the F1-score for both models over the baseline score.

### 4.5. Classifier Two-Sample Test

To perform the Classifier Two-Sample Test (CTST), we utilized a simple neural network. For comparison, an LSTM network was used. The neural network consisted of one layer with 20 units and a ReLU activation function followed by a sigmoid activation [40]. The LSTM consisted of 32 units followed by a sigmoid activation function. Both binary classifiers were trained for 100 epochs and leveraged the Adam optimizer with a learning rate of 0.001. We generated a dataset with our trained LSTM generator, which consisted of MOS and non-MOS sequences based on a fixed random seed. The samples from the synthetic physiological measurements in the dataset had the same cardinality as the real samples from physiological dataset. The training dataset consisted of 196 synthetic and real MOS, as well as 1026 synthetic and real non-MOS sequences, resulting in a total of 2444 sequences. The hold-out test dataset consisted of 84 MOS and 256 non-MOS, which resulted in 680 in total. After training, the classifier’s accuracy was evaluated on the hold-out test dataset. The results are shown in Table 4. As can be observed, for both classification models, the accuracy of our implementation was near the chance level, which means that our cGAN captured important features from the real distribution. Once our implementation reached the lowest CTST score, the score fluctuated in the range between 0.7 and 0.5903, which was caused by the random space and the known insatiability of the GANs. These results suggest that the generated data distribution learned the real data distribution.

## 5. Discussion and Future Research

The proposed cGAN can capture the time–frequency features of the GSR and ST signals and, therefore, produce samples that are indistinguishable from real samples. This includes samples following a similar distribution and and new samples that can fill gaps in the real dataset, which explains the improvement of the classifier score on our stress dataset. However, training a GAN on a small and imbalanced dataset is not trivial. It takes a considerable amount of work to find the optimal hyperparameters. At the same time, the instability of the GAN training makes it difficult to pinpoint the best hyperparameters. In addition, the GAN may produce only one category of samples, which is called mode collapse in the literature. However, this effect was notably reduced in our work by using a cGAN with a diversity term. Summarizing all these points, one can see that there is still much research to be performed on GAN training, especially for small physiological time series data. This is particularly interesting due to the great potential that this method has shown, both in the present study and in the literature. In future works, it would be interesting to see new architectures or other loss functions such as the Wasserstein GAN [44] on physiological measurement dataset.

In our research, we utilized an FCN discriminator to generate physiological time series data. We chose this architecture because it outperformed other state-of-the-art techniques in time series data generation in our experiments on our dataset and, in recent years, time series classification with CNN architectures have become more popular [10]. However, it would also be interesting to evaluate our architecture on different datasets or to try to utilize a residual network [45] as a discriminator, which has become state-of-the-art in many time series classification tasks.

Our self-designed data acquisition campaign in the controlled laboratory environment provided us with the physiological stress data for this work. In this campaign, we triggered ten MOSs per participant. These MOSs were used for the stress label in the cGAN and in the classifier. However, with these samples, it can happen that an MOS falls into the non-MOS category because the participants can also experience a stress moment between the stressors due to various factors, which is reflected in the physiological data. This false classification influences the sequences produced by our cGAN since the distributions for the labels are delimited less strongly. To overcome this issue, in the future, two possible approaches could be investigated: a first possible way could be improving the experimental protocol or, secondly, using active learning [46] to relabel noisy sequences by experts. In this scenario, it would also be possible to go beyond binary classification and introduce more granular levels of labels such as strong MOS, medium MOS, and non-MOS.

In this work, we used a data preprocessing schema to resample the physiological signal to 16 s sequences, which reduced the complexity of the whole dataset and significantly improved our results. However, the main drawback of this approach is that participants’ individuality in terms of stress reaction sequences is lost. Since the literature [1,47] has identified that stress reactions are subjective and people react differently, an interesting approach for the future would be to use the latent space in the cGAN to generate individual MOS and non-MOS per participant.

As described earlier, we only used GSR and ST as our stress indicators in this study and achieved good results on our test data. A major advantage of using only two physiological signals is that it is easily transferable to real scenarios and everything can be recorded with just one wearable. Nevertheless, the integration of heart rate variability and heart rate could further improve the accuracy of our classifier.

## 6. Conclusions

In this work, we proposed a data augmentation technique leveraging a cGAN to detect Moments Of Stress (MOS) with different neural network classifiers on a small and imbalanced physiological measurement dataset. Additionally, we proposed an LSTM-FCN cGAN architecture combined with a diversity term. For this reason, we measured GSR and ST using a low-cost wearable sensor. In a controlled laboratory data acquisition campaign, we generated ground truth data for moments of stress via audio stimuli, which we used as a labeled physiological dataset to train our classifiers. We used a combination of data preprocessing and cGAN data augmentation, to tackle the problem of training deep learning algorithms on a small and imbalanced dataset. With the help of synthetic samples from the cGAN, we were able to detect 72.62% of the induced MOSs on the testing dataset and, therefore, improved the performance of two different classifier models in terms of recall and F1-score. During the experiments, the LSTM model performed particularly well with an improvement in recall around 19.05% and 11.03% in the F1-score, compared to the baseline LSTM classifier score.

Concluding, the research presented in this paper showed that it is possible to train a cGAN on a small and imbalanced physiological time series dataset collected with wearable sensors. Furthermore, it showed that these synthetic data can be used to train a classifier on moments of stress, which improved the overall performance. The architecture that was tested in our experiments is described in Section 3 and can be used to enhance classification performance where time series physiological data are used.

## Figures and Tables

**Figure 1 sensors-22-05969-f001:**
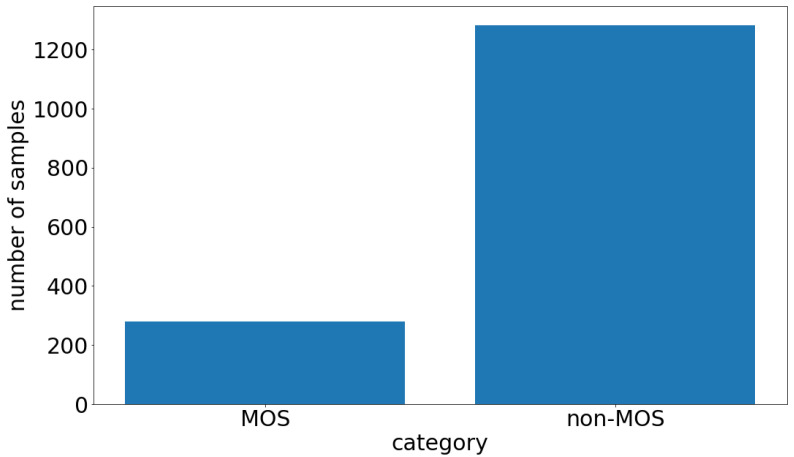
The label distribution of our physiological measurement dataset. The left bar is the Moment Of Stress (MOS) class, and the right one is the non-MOS class.

**Figure 2 sensors-22-05969-f002:**
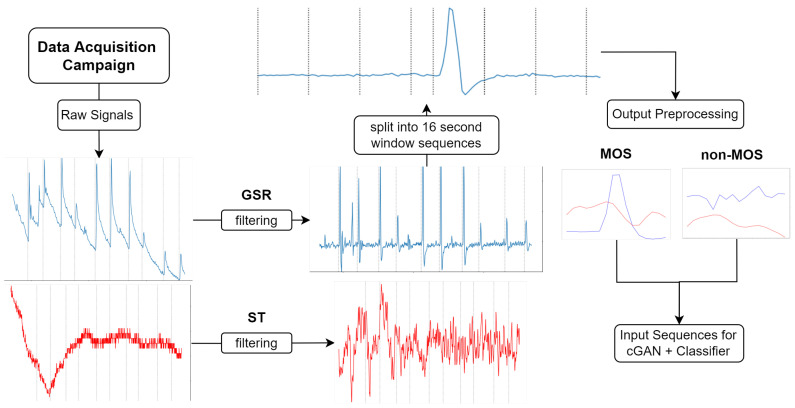
Prepare and preprocess raw signals for the cGAN and the stress classifier. The red line indicates ST and the blue line indicates GSR. The dotted line in the raw signals and in the filtered signals indicates induced MOS. In the window plot the dotted line indicates split index.

**Figure 3 sensors-22-05969-f003:**
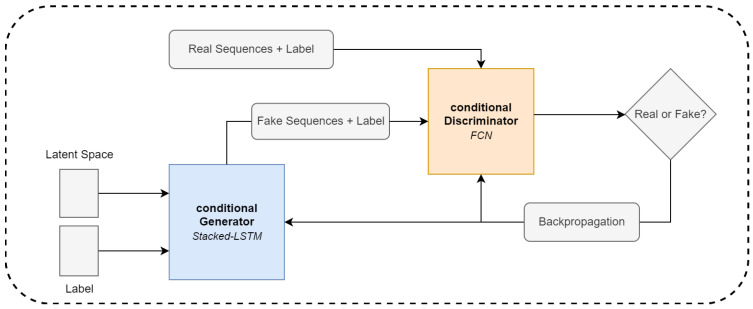
cGAN workflow.

**Figure 4 sensors-22-05969-f004:**
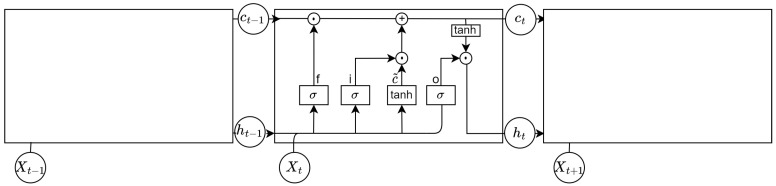
LSTM cell.

**Figure 5 sensors-22-05969-f005:**
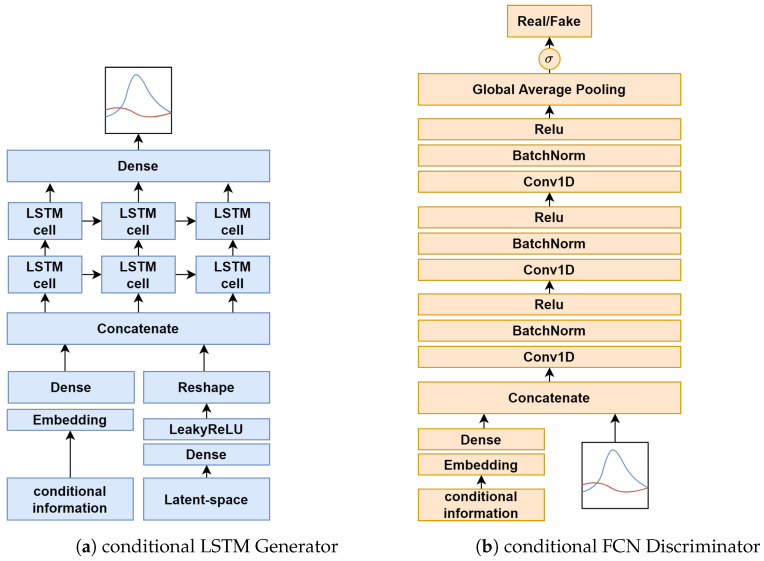
The architecture of our conditional GAN. In the input and output figure in (**a**,**b**), the blue line indicates GSR and the red line indicates ST, which shows a prototypical MOS.

**Figure 6 sensors-22-05969-f006:**
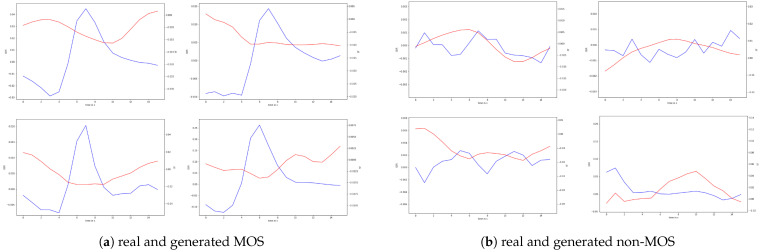
Visual comparison of real and generated samples. The red line shows a standardized and filtered 16 s ST signal. The blue line shows a standardized and filtered 16 s GSR signal. There are always two generated and two real signal samples arranged in a 2 × 2 grid.

**Figure 7 sensors-22-05969-f007:**
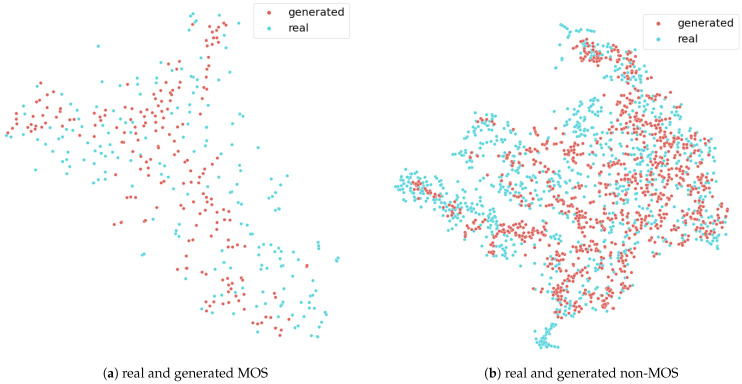
The two figures show the results from the t-sne. The red points are the generated points, and the blue points are the real data points.

**Table 1 sensors-22-05969-t001:** The results from the classifier experiments are shown. The different scores indicate the best possible results we reached during training.

**FCN**	**Recall**	**Precision**	**F1**	**Accuracy**
baseline	0.4881	0.8542	0.6212	0.84
RCGAN TGTR	0.5357	0.7377	0.6207	0.8382
RCGAN DAug	0.5833	0.7903	0.6712	0.8588
TimeGAN TGTR	0.5833	0.6203	0.6012	0.8088
TimeGAN DAug	0.6429	0.71	0.6750	0.8471
Ours TGTR	0.5238	0.7719	0.6241	0.84
Ours DAug	0.7262	0.7439	0.7349	0.8676
**LSTM**	**Recall**	**Precision**	**F1**	**Accuracy**
baseline	0.5357	0.8654	0.6618	0.8647
RCGAN TGTR	0.4762	0.6250	0.5405	0.8000
RCGAN DAug	0.6190	0.7324	0.6709	0.8500
TimeGAN TGTR	0.5833	0.6533	0.6163	0.8206
TimeGAN DAug	0.5952	0.8065	0.6849	0.8647
Ours TGTR	0.6786	0.7600	0.7170	0.8618
Ours DAug	0.7262	0.8243	0.7721	0.88

**Table 2 sensors-22-05969-t002:** The results of the classification between real and generated performed by experts. The accuracy score is the mean of the participants’ performance.

	Accuracy
Real/Generated	0.4575

**Table 3 sensors-22-05969-t003:** The binary classification of physiological measurement data according to stress moments performed by experts.

	Recall	Precision	F1	Accuracy
All Sequences	0.7567	0.7814	0.7487	0.8175
Real	0.74	0.7019	0.6973	0.765
Generated	0.7733	0.8816	0.8065	0.870

**Table 4 sensors-22-05969-t004:** Results of the classifier two-sample test. The closer to the chance level, the better are the results.

	Neural Net	LSTM
CTST LSTM-FCN	0.6221	0.5903

## Data Availability

Not applicable.

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
