# Peer review of "A Conditional GAN for Generating Time Series Data for Stress Detection in Wearable Physiological Sensor Data"

_sensors, 2022, doi:10.3390/s22165969_

Round 1
Reviewer 1 Report
In this manuscript, the authors combined the LSTM and the FCN into the cGAN to generate synthetic physiological data. Those synthetic physiological data could be used as training samples to establish an accurate stress detector.
The structure of this paper is apparent; the authors introduced the problem statements first and then came to the literature review, which makes readers understand why the study should be accomplished. The English is well written and straightforward. The analysis steps are apparent, and the results are well explained.
I have a positive attitude to this research, and only a few issues should be treated.
1. Why do the authors combine LSTM and FCN into the cGAN, not the other neural network structures? There are limited studies applying the cGAN to augment datasets in body stress detection. However, many similar researches have been conducted in time-series data argumentation. Thus, the logic of selecting the two specific network structures (i.e., LSTM and FCN) should be well explained and discussed.
2. Since one key highlight of this manuscript is “proposing” a new GAN structure. Ablation and comparative studies are two key issues that should be further discussed. If the state-of-the-art method in a similar field (i.e., time-series data argumentation) is used as a benchmark and compared with the proposed method. The quality of the manuscript would be highly improved.
3. All Figures, please improve the quality and font size to be more visible. Please improve the graphic quality (resolution).
Please adjust the formation of Figure 5. In its current form, it is really hard to read.
Some typos:
4. line 47, should be Figure 1, not 1.
5 line 123, check the position of the reference number [15]
6. The figure captions should below the figure, not above.
Reviewer 2 Report
A very well written manuscript. A successful and significative implementation of GANs which generates synthetic physiological signals such as galvanic skin response (GSR) and skin temperature (ST) for a classification task.
Some minor comments:
· - Each input window of 16 seconds of the signal after preprocessing and before feeding the model downsampled at 1 Hz has 16 values. Is that the input to the LSTM generator and the FCN discriminator or more sequential windows have been used as input at each forward pass? The LSTM needs more than one 16 sec sequences as input segment and the FCN needs larger input if kernel sizes {8,5,3} are to be used. Please clarify the dimension of the input and the output of the generator and consequently the input for the discriminator. In line 264, the X matrix is mentioned to have dimensions n x t x d (size of samples x length of sequence x number of features).What do you mean by features and length of sequence? Please clarify.
· - What is the finally chosen value of λ hyperparameter in the expression (10)
· - The label information is pushed through an embedding layer. What kind of embedding? Please give some more information.
· - The tables 2 and 3 could be more informative if the scores for accuracy were shown.
- GAN’s are known to be unstable around their equilibrium point. Could you discuss a little bit more this point with regard to your implementation. Was it easy for the proposed model to reach the stationary point of equilibrium that every GAN theoretically has? Did you monitor the way to this equilibrium? How does the proposed c-GAN behave in terms of approaching the equilibrium?
